# A New Perspective on Regenerative Medicine: Plant-Derived Extracellular Vesicles

**DOI:** 10.3390/biom15081095

**Published:** 2025-07-28

**Authors:** Yuan Zuo, Jinying Zhang, Bo Sun, Xinxing Wang, Ruiying Wang, Shuo Tian, Mingsan Miao

**Affiliations:** Academy of Traditional Chinese Medicine, Henan University of Chinese Medicine, Zhengzhou 450046, China; zzuoyuan@163.com (Y.Z.); zhang149162@163.com (J.Z.); 13466144534@163.com (B.S.); 18836945424@163.com (X.W.); 13663811635@163.com (R.W.); tianshuo0416@163.com (S.T.)

**Keywords:** plant-derived extracellular vesicles, extraction and isolation, regenerative properties, skeletal muscular system disorders, neurodegenerative diseases, tissue repair

## Abstract

Plant-derived extracellular vesicles (PDEVs) are nanoscale, phospholipid bilayer-enclosed vesicles secreted by living cells through cytokinesis under physiological and pathological conditions. Owing to their high biocompatibility and stability, PDEVs have attracted considerable interest in regenerative medicine applications. They can exhibit the capacity to enhance cellular proliferation, migration, and multi-lineage differentiation through immunomodulation, anti-inflammation effects, antioxidative protection, and tissue regeneration mechanisms. Given their abundant availability, favorable safety profile, and low immunogenicity risks, PDEVs have been successfully employed in therapeutic interventions for skeletal muscle disorders, cardiovascular diseases, neurodegenerative conditions, and tissue regeneration applications. This review mainly provides a comprehensive overview of PDEVs, systematically examining their biological properties, standardized isolation and characterization methodologies, preservation techniques, and current applications in regenerative medicine. Furthermore, we critically discuss future research directions and clinical translation potential, aiming to facilitate the advancement of PDEV-based therapeutic strategies.

## 1. Introduction

Extracellular vesicles (EVs) are lipid bilayer-delimited spherical nanoparticles secreted by both prokaryotic and eukaryotic cells. Initially identified in 1983, these membrane-bound vesicles typically exhibit a diameter ranging from 30 to 150 nm [1]. They encapsulate various bioactive molecules including nucleic acids, proteins, and other cellular components that reflect the molecular signature of their parent cells. These molecular cargos enable EVs to functionally mimic their cell of origin and participate in complex intercellular signaling pathways. As evolutionarily conserved nanoscale messengers, EVs play a pivotal role in mediating intercellular communication by facilitating the transfer of biological information and exchange of macromolecules not only between neighboring cells but also across different species [2]. Due to their remarkable bioactivity and therapeutic efficacy, EVs have emerged as a focus of multidisciplinary research, with significant applications in drug delivery platforms, disease treatment strategies, and clinical diagnostic development. Particularly in regenerative medicine, EVs have demonstrated considerable potential for treating diverse pathological conditions, including musculoskeletal disorders, cardiovascular diseases, neurodegenerative pathologies, and various tissue regeneration processes. For example, Cosenza et al. [3] demonstrated that intra-articular administration of bone marrow mesenchymal stem cell-derived EVs (BMSC-EVs) in a murine osteoarthritis (OA) model significantly attenuated joint degeneration, suggesting their chondroprotective potential. They have been successfully isolated from diverse biological sources, including bacterial, mammalian, and plant cellular systems [4]. Notably, plant-derived extracellular vesicles (PDEVs) are similar to mammalian-derived extracellular vesicles (MDEVs) in terms of biogenetics and morphology [5]. PDEVs are abundant, highly productive, low-cost to produce, and possess a relatively low immunogenicity due to no carrying of zoonotic pathogens [6], and have been widely used in the field of regenerative medicine. This review mainly provides a comprehensive overview of PDEVs, systematically examining their biological properties, standardized isolation and characterization methodologies, preservation techniques, and current applications in regenerative medicine. Furthermore, we critically discuss future research directions and clinical translation potential, aiming to facilitate the advancement of PDEV-based therapeutic strategies. (See Figure 1).

## 2. Biological Characterization of PDEVs

### 2.1. Biogenesis Pathways of PDEVs

According to the current research, there are three primary biogenesis pathways of PDEVs: (1) the exocyst-positive organelle (EXPO)-mediated pathway, (2) multivesicular body (MVB)-dependent vesicle formation, and (3) vacuole-mediated secretion [7]. Among them, the MVB pathway is recognized as the predominant mechanism for PDEV biogenesis [8]. The presence of MVBs in plant cells was first confirmed in 1967. Subsequent studies have demonstrated that PDEVs are secreted via MVB–plasma membrane fusion, a conserved eukaryotic mechanism analogous to exosome release in mammalian systems. An et al. [9] unequivocally identified MVBs in Hordeum vulgare leaf cells during fungal infection using transmission electron microscopy (TEM). MVBs, classified as late endosomal compartments or pre-vacuolar structures, typically measure 200–1000 nm in diameter and contain numerous intraluminal vesicles (ILVs) ranging from 30–100 nm in size [10]. There are two fates for MVBs: one is to fuse with vacuoles or lysosomes, mediating the degradation or recycling of cargo molecules within the cell [11]. The other is fusion with the plasma membrane, leading to the release of their contents, including ILVs, into the extracellular environment, and the released vesicles are called exosomes [12]. An alternative biogenesis pathway for PDEVs is EXPO. It is functionally associated with Exo70E2. Exo70E2 is a homolog of the yeast and animal exocyst protein Exo70, and it has been identified in Arabidopsis thaliana and Nicotiana tabacum suspension cells. EXO70 is a key subunit of the secretory complex. Previous studies in yeast, animals, and plants have shown that the secretory complex mediates the tethering process between post-Golgi vesicles and the plasma membrane [12,13]. EXPO is different from MVB and does not belong to the endocytic pathway. It is considered to be similar to the formation of autophagosomes. Beyond MVBs and EXPO, vacuoles represent a third potential source of PDEVs through two interconnected mechanisms: (1) MVB–vacuole fusion followed by subsequent vacuole–plasma membrane fusion, releasing intraluminal vesicles (ILVs) into the extracellular space [14], and (2) direct vacuole–plasma membrane fusion. Supporting this model, infection of Arabidopsis thaliana with Pseudomonas syringae triggers tonoplast–plasma membrane fusion in leaf epidermal cells, resulting in extracellular release of vacuolar contents, including proteolytic enzymes. This demonstrates the plasticity of vacuolar membrane trafficking under stress conditions [15] (see Figure 2 for details). The biogenesis of exosomes initiates with ILV formation within MVBs, a process governed by two coordinated mechanisms. Firstly, the endosomal membrane reorganization is mediated by tetraspanins (particularly CD9 and CD63), which show significant enrichment in developing ILVs [16]. Secondly, the Endosomal Sorting Complexes Required for Transport (ESCRTs) are recruited to the site of ILV formation [17]. Four different types of ESCRTs have already been identified: 0, I, II, and III. ESCRT 0 can recognize ubiquitinated proteins on the outside of endosomal membranes [18]. ESCRT I and II are recruited to the cytoplasmic side of early endosomes by various stimuli. ESCRT I binds ubiquitinated cargo with ESCRT II-activated endosomes. ESCRT III assembles through programmed cell death 6-interacting protein (PDCD6IP or ALIX), and, as part of the ESCRT I complex, it binds with tumor susceptibility gene 101 [19]. ALIX acts as an intermediary in the interaction between ESCRT I and ESCRT III, as it binds to the TSG101 component of ESCRT I and the charged MVB protein 4A of ESCRT III. The process is accomplished by chelating MVB proteins and recruiting deubiquitinating enzymes that remove the ubiquitin tags from cargo proteins before sorting the reproduced proteins into ILVs. Finally, ESCRT III is disassembled for recycling by the AAA-ATPase inhibitor potassium transport growth defective 1 protein [20]. In summary, PDEVs are primarily released into the extracellular space through the fusion of MVBs and EXPOs with the plasma membrane. Additionally, the fusion of MVBs/EXPOs with vacuoles and autophagosomes may also be involved in the formation of PDEVs.

### 2.2. Classification of PDEVs

PDEVs are nanosized vesicles with a phospholipid bilayer released by cells via exocytosis under both physiological and pathological conditions. The study of PDEVs originated in 1965 when Jensen first observed granular “single-membrane spheres” in the extracellular space of cotton cells using transmission electron microscopy, marking a pivotal discovery in this field [21]. PDEVs encapsulate a variety of bioactive molecules that mediate intercellular and interspecies communication by facilitating the transfer of biological information and cargo. They have a regulatory role in multiple physiological and pathological processes, including immune modulation, cellular proliferation, apoptosis, and angiogenesis [15]. They can be classified according to their physical properties, biological origin, and biochemical composition, and can be divided into exosomes, microvesicles, and apoptotic bodies. Exosomes (30–100 nm) mainly originate from vesicles secreted by intracellular membrane structures within organelles. Microvesicles (50–1000 nm or 100–1000 nm) originate from the plasma membrane, while apoptotic vesicles (50–5000 nm) are produced within apoptotic cells [22], as shown Figure 3.

### 2.3. Physicochemical Characterization of PDEVs

#### 2.3.1. Physical Characterization of PDEVs

In accordance with the guidelines established by the International Society for Extracellular Vesicles (ISEV), comprehensive characterization of plant-derived extracellular vesicles (PDEVs) should include (1) morphological analysis; (2) quantitative assessment of particle number and concentration; (3) particle size distribution profiling; (4) quantification of total protein, lipid, and RNA content; (5) proteomic composition analysis; (6) identification of non-protein markers; and (7) evaluation of other functionally relevant constituents [23]. The morphology characterization of PDEVs is typically performed using electron microscopy techniques, including scanning electron microscopy, transmission electron microscopy, cryo-electron microscopy, and atomic force microscopy. Among these techniques, transmission electron microscopy offers superior resolution compared to scanning electron microscopy, enabling the visualization of finer structural details with greater clarity. However, both methods require the samples to be dehydrated and fixed, which can cause PDEVs to deform, resulting in a teacup or cup-like appearance under the microscope [24]. In contrast, cryo-electron microscopy does not require fixation or staining, allowing for more authentic images of PDEVs, with shapes close to spherical [25]. The commonly used methods to measure the size and surface charge of PDEVs are dynamic light scattering and nanoparticle tracking analysis. Dynamic light scattering, due to its low resolution and inability to measure particle concentration, has gradually been replaced by nanoparticle tracking analysis technology. Characterization of various PDEV preparations has revealed consistently negative zeta potentials, with values ranging from −1.5 to −49.2 mV across different samples.

#### 2.3.2. Biochemical Characterization of PDEVs

PDEVs are nanoscale lipid bilayer-enclosed vesicles. They encapsulate lipids, proteins, nucleic acids, metabolites, and other molecules. Lipids constitute the primary structural components of the PDEV bilayer. Gel electrophoresis and high-throughput sequencing analyses can be employed for their characterization. For instance, glucosylceramide has been identified as the predominant lipid species in Aloe vera-derived EVs, where it plays a crucial role in facilitating membrane curvature generation, thereby promoting PDEV formation and secretion [26]. PDEVs predominantly contain functional proteins associated with cytoskeleton organization, metabolic signaling, intracellular transport, and secretory pathways. Researchers can choose methods such as gas chromatography–mass spectrometry, liquid chromatography–mass spectrometry, Western blotting, the Coomassie Brilliant Blue method, the Bicinchoninic Acid Assay (BCA) method, and proteomics to detect protein components. For instance, the proteins in EVs derived from ginseng and turmeric exhibit significant correspondence with proteins implicated in diverse biological processes, molecular functions, and cellular components. These PDEVs encapsulate a heterogeneous population of nucleic acid, including genomic DNA, microRNAs (miRNAs), and small interfering RNAs (siRNAs). Notably, these nucleic acid species represent a promising new generation of therapeutic agents with substantial potential for the treatment of various pathological conditions. Li et al. [27] artificially synthesized lipid complexes carrying sRNAs by mimicking the structure of EVs, demonstrating that sRNA-6 from dandelion-derived EVs and sRNA-m7 from Rhodiola-derived EVs can be effectively delivered to cells.

## 3. Extraction, Isolation, and Storage of PDEVs

PDEVs have similar functions to MDEVs, but due to the complexity of the biological fluid, contamination with non-vesicular components such as lipoprotein and nucleoprotein complexes often occurs [28]. As a result, the purification steps for MDEVs pose obstacles to separating them from the supernatant of tissue cultures. Therefore, PDEVs are becoming more attractive because they can overcome these defects. (1) The extraction of PDEVs is simpler and more cost-effective compared to MDEVs, as it does not rely on complex cell culture systems. Moreover, (2) PDEVs can be readily obtained from abundant plant sources, significantly reducing production costs. (3) Another key advantage is their lower risk of pathogen contamination, enhancing their safety profile. (4) Furthermore, PDEVs exhibit strong potential for scalable manufacturing, making them a promising candidate for industrial and biomedical applications. PDEVs can be extracted from plant tissues through mechanical disruption methods such as juicing or extrusion. Currently, multiple isolation techniques are employed to purify PDEVs, including ultracentrifugation, density gradient centrifugation, ultrafiltration, size exclusion chromatography, polymer precipitation, tangential flow filtration, and asymmetric flow field-flow fractionation. Furthermore, emerging technologies such as microfluidic systems, nano-flow cytometry, and nanoscale lateral displacement array are increasingly being utilized for high-precision isolation and characterization of PDEVs. Ultracentrifugation is simple to perform and inexpensive since it does not require solvents or substances. However, this process is too time-consuming and requires expensive machinery [29]. Ultrafiltration is a size-based technique involving membrane filters and pressure, resulting in faster separation, and does not necessitate special equipment. The efficiency of this method is greater than that of ultracentrifugation, and the time required is shorter [30]. Size exclusion chromatography is a simple and economical method. Working with gravity does not affect the integrity of the PDEVs and results in great reproducibility; however, specific equipment is needed, its scalability is difficult, and it takes a long time to perform [31]. Polymer precipitation is easy to perform, cost-effective, and does not affect the quality of the exosomes. However, due to vesicle aggregation, the process can result in co-isolation with other PDEVs or other cellular components [32], as illustrated in Table 1 and Figure 4.

As a novel class of therapeutic agents or delivery vehicles, PDEVs require optimized storage conditions to preserve their structural integrity and biological functionality. Studies indicate that PDEVs remain stable for approximately one year when stored at –80 °C, whereas storage at –20 °C maintains stability for up to three months [33]. Leng Y et al. [34] found that the optimal storage temperatures for blueberry-derived EVs were 4 °C for short-term storage and −80 °C for long-term storage. Storage at 4 °C helped to prevent ice crystals from damaging the phospholipid bilayer membranes of the PDEVs, while storage at −80 °C slowed down the rate of degradation and maintained the particle morphology. Richter M et al. [35] found that PDEVs stored at −80 °C and 4 °C had higher particle recovery than freeze-dried PDEVs.
biomolecules-15-01095-t001_Table 1Table 1Common extraction and isolation methods for PDEVs.Separation TechnologyPrincipleAdvantagesDisadvantagesReferencesUltracentrifugationParticle size and densityHigh purity, simple operation, suitable for the extraction of most PDEVsCumbersome, time-consuming, requires specialized ultra-high-speed centrifuges and centrifuge tubes, costly[36]Polymer precipitationSolubilityHigh output, simple operation, low costLow purity, may have residual impurities[37]UltrafiltrationParticle sizeSimple, low-cost, and time-consuming methodLow purity and pressure may cause damage to PDEVs, susceptible to irreversible clogging of macromolecules[38]Size-exclusion chromatographyParticle sizeHigher purity, more structural integrity, more complete retention of relevant physical properties and biological activityLow output[39]Tangential flow filtrationParticle sizeHigh purity and throughput, suitable for concentration and separation of large samples; more structural integrity of PDEVsHigh cost[40]Immunomagnetic bead methodThe specific binding between antibody and exosome-specific markerHigh outputHigh-cost and easy-to-destroy exosome structure[41]MicrofluidicsDifferences in physical and chemical propertiesHigh speed, high sensitivity, low costLower purity, only suitable for the detection of biological indicators[42]

## 4. Applications of PDEVs in Regenerative Medicine

Regenerative medicine aims to restore the structure and function of damaged tissues or organs. From a regenerative medicine standpoint, PDEVs exhibit considerable therapeutic potential by addressing key limitations inherent to cell-based therapies. Notably, PDEVs can be functionally engineered through multiple modification strategies to achieve targeted therapeutic outcomes. Compared with conventional chemical drugs, PDEVs demonstrate superior transdermal delivery capabilities, including (1) enhanced cellular fusion with skin cells, (2) efficient drug transport across the stratum corneum without disrupting skin integrity, and (3) formation of sustained drug reservoirs both on and within the skin surface. These unique properties enable localized, prolonged drug release for optimized therapeutic efficacy. First of all, sugar beet-derived EVs can antioxidize and scavenge necrotic tissues, showing great potential in skin repair and delaying aging. They can promote the formation of new intima in vascular endothelial cells and increase the production of collagen I, collagen III, and hyaluronan synthase in fibroblasts [43]. Next, EVs derived from ginseng, pine needles, and sage can stimulate hair follicle regeneration. Finally, ginger-derived EVs can promote intestinal wound recovery by affecting the expression of mitochondrial and cytoplasmic proteins. PDEVs contain a variety of bioactive components and are carriers of communication between cells and their surroundings. They play an important role in the prevention and treatment of degenerative diseases. For instance, strawberry-derived EVs can be taken up and internalized by human mesenchymal stromal cells. They are rich in vitamin C and have significant antioxidant activity, playing a role in reducing the risk of chronic aging and degenerative diseases [44]. Among other features, in bone-related degenerative diseases, ginseng-derived EVs exhibit higher vitality and proliferation capacity of bone marrow-derived macrophages while also inhibiting the formation of osteoclasts. Moreover, in a mouse model of lipopolysaccharide-induced bone resorption, ginseng-derived EVs inhibited the differentiation of osteoclasts. The higher ratios of ginsenosides Rb1 and Rg1 contained in these EVs are more effective in inhibiting osteoclast differentiation than ginsenosides Rb1 or Rg1 alone. Therefore, these PDEVs have potential value in the clinical prevention and treatment of osteoporosis and other bone loss diseases [45]. Ginseng is also commonly used in the treatment of neurodegenerative diseases. Ginseng-derived EVs can exert neuroprotective effects by maintaining homeostasis and participating in anti-inflammatory, antioxidant, and anti-apoptotic cell death processes. They hold great potential in treating brain diseases such as stroke and Parkinson’s disease [6]. Over the past decades, PDEVs have emerged as novel drugs for the treatment of various diseases and conditions due to their multifaceted biological functions.

### 4.1. Application of PDEVs in Promoting Wound Healing and Tissue Repair

Tissue damage and aging contribute to functional impairment, structural deformity, and traumatic injury. Regenerative medicine aims to restore compromised physiological functions through the repair, replacement, or regeneration of affected cells, tissues, and organs. A critical aspect of this process involves establishing a regenerative microenvironment capable of both mitigating external stressors and promoting stem cell differentiation. PDEVs are natural bioactive lipid bilayer nanovesicles containing proteins, lipids, ribonucleic acid, and metabolites. They have great potential in promoting cell growth, migration differentiation, and various types of tissue repair. PDEVs possess bioactivities such as immunomodulation, microbiota regulation, antioxidant, and anti-aging, and are highly valuable in resisting external stimuli and promoting tissue repair. Saponaria-derived EVs promote enhanced vessel formation in HUVEC by decreasing the expression of pro-inflammatory genes such as interleukin-6 and interleukin-1β [46]. *Momordica charantia*-derived EVs promote cell proliferation, inhibit apoptosis, and attenuate DNA damage in irradiated (16 Gy, X-rays) H9C2 cells, promoting tissue repair [47]. Ginger-derived EVs stimulate the Wnt/β-catenin signaling pathway, thereby increasing the population of intestinal epithelial stem cells [48]. The bioactive substances in PDEVs can induce cell proliferation and differentiation, thereby accelerating the tissue repair process. For example, these EVs can stimulate osteoblast proliferation and migration, thereby enhancing tissue repair and regeneration. Additionally, they exhibit immunomodulatory properties by suppressing the release of pro-inflammatory cytokines, which mitigates inflammation-mediated secondary tissue damage and accelerates wound healing processes. Angiogenesis is an important process of tissue repair and wound healing (as depicted in Table 2 and Figure 5). PDEVs exhibit significant therapeutic potential in regenerative medicine through their pro-angiogenic properties, which enhance neovascularization to improve oxygen supply and accelerate wound healing. As a novel class of natural therapeutics, PDEVs demonstrate favorable biosafety profiles with low immunogenicity, making them promising candidates for treating diverse tissue injuries and degenerative diseases. Their regenerative capabilities extend to promoting cellular proliferation and differentiation, as well as facilitating vascular and neural regeneration, thereby supporting the development of functional tissue constructs with broad clinical applications.

#### 4.1.1. Musculoskeletal System Diseases

The human musculoskeletal system, a biomechanically integrated network, comprises osseous structures organized into the skeleton, striated musculature, and associated connective tissues including tendons, ligaments, articular cartilage, and synovial joints. Pathological alterations in this system arise from multifactorial interactions among progressive degenerative processes (e.g., age-related extracellular matrix degradation), metabolic dysregulation (e.g., calcium-phosphate homeostasis imbalance), and biomechanical overloading. These etiological determinants frequently manifest as clinically defined degenerative disorders, with OA (characterized by articular cartilage erosion), osteoporosis (marked by trabecular bone loss), and sarcopenia (exhibiting type II muscle fiber atrophy) representing prevalent nosological entities. Currently, the treatment of disorders of the musculoskeletal system, especially degenerative diseases, is characterized by significant limitations and adverse effects, whether through pharmacologic or surgical interventions [49]. EVs have emerged as promising biopharmaceutical agents with significant therapeutic potential. Specific human-derived EVs, particularly those isolated from peripheral blood and mesenchymal stromal cells (MSCs), are effectively internalized by joint-resident cells including chondrocytes, synovial fibroblasts, and osteoblasts. Mechanistic studies have revealed that these EVs exert multifaceted therapeutic effects through three primary pathways: (1) immunomodulation via regulation of both innate and adaptive immune responses; (2) restoration of cellular motility and functional activation through cytoskeletal reorganization; and (3) attenuation of senescence-associated secretory phenotype (SASP) via modulation of the p53/p21 signaling axis. PDEVs have similar morphological structures and biological activities to MDEVs and have been shown to possess excellent antioxidant, anti-inflammatory, cell proliferation, differentiation, and enhanced migration as well as apoptosis reduction and angiogenesis promotion properties. They mainly target osteoblast lineage, fibroblasts, chondrocytes, etc., thus regulating proliferation and differentiation, and promoting cell migration [49], and have a promising application in the treatment of musculoskeletal system diseases.

##### Osteoarthritis

Osteoarthritis represents a chronic degenerative joint disease characterized by progressive articular cartilage deterioration and pathological bone remodeling. The disease pathogenesis involves multiple interrelated pathological processes: synovial inflammation, progressive loss of articular cartilage, degeneration and denaturation of meniscus and ligaments, thickening of the subchondral bone, and formation of bony encumbrances [50]. The development of OA is strongly associated with several well-established risk factors, including obesity, advanced age, congenital joint malformations, joint trauma history, repeated stress, and overuse. Due to the extremely poor ability of cartilage regeneration on the joint surface, once cartilage damage occurs, it initiates a pathological cascade leading to progressive joint dysfunction, irreversible cartilage degradation, and secondary joint complications. Current therapeutic strategies for OA utilizing PDEVs primarily target four key pathological aspects: chondrocyte repair, anti-inflammatory effects, oxidative stress reduction, and extracellular matrix regulation. Accumulating evidence has demonstrated the chondroprotective potential of PDEVs. Chen et al. [51] developed an innovative therapeutic platform by encapsulating an independent and controllable nanosized plant-derived photosynthetic system based on nanothylakoid units within chondrocyte-specific membranes. This light-responsive system demonstrated dual functionality: localized enhancement of intracellular ATP and NADPH production upon photostimulation, and significant improvement in the anabolic metabolism of degenerated chondrocytes. Importantly, this approach effectively corrected energy metabolism dysregulation at the cellular level, thereby restoring metabolic homeostasis in chondrocytes. Such metabolic modulation not only maintains cartilage homeostasis but also shows promising potential for halting OA progression. Yıldırım M et al. [52] demonstrated that tomato-derived EVs exhibit chondroprotective and reparative effects on chondrocytes. Specifically, these EVs upregulated the expression of critical chondrogenic markers, including ACAN, SOX9, and COMP, as well as essential extracellular matrix proteins such as type II collagen COL2 and type XI collagen COLXI. Furthermore, they enhanced the chondrogenic differentiation potential of human adipose-derived mesenchymal stem cells (MSCs), suggesting their therapeutic utility in cartilage regeneration. In another study, EVs derived from curcumin-treated MSCs were shown to modulate key signaling pathways in OA [53]. These EVs significantly increased the expression of miR-143 and miR-124 while suppressing NF-κB and ROCK1, thereby mitigating OA progression. These findings highlight the potential of MSC-derived EVs in treating degenerative cartilage diseases. Given their diverse biological sources, high biocompatibility, and enhanced ability to traverse biological barriers, PDEVs represent a promising therapeutic strategy for osteoarthritis and other cartilage-related disorders.

##### Osteoporosis

Osteoporosis (OP) is a systemic skeletal disorder characterized by reduced bone mineral density and deterioration of bone microarchitecture, resulting in increased bone fragility and elevated fracture risk. The pathogenesis of OP primarily stems from an imbalance between osteoblastic bone formation and osteoclastic bone resorption, leading to progressive net bone loss. Key risk factors include aging, prolonged glucocorticoid therapy, and excessive alcohol consumption, which disrupt bone remodeling homeostasis [54]. Currently, the primary pharmacological treatments for OP include selective estrogen receptor modulators (SERMs) (e.g., Raloxifene), RANKL inhibitors (e.g., Denosumab), and parathyroid hormone analogs (e.g., Teriparatide and Abaloparatide). Although these drugs demonstrate clinical efficacy in managing bone loss, their long-term use is often limited by adverse effects and the potential development of drug resistance [55]. Consequently, there is an urgent need to explore novel therapeutic approaches with improved safety profiles and sustained effectiveness. Seo K et al. [45] identified that ginseng-derived EVs (GDEVs) significantly suppress RANKL-induced osteoclastogenesis through modulation of key signaling pathways. Specifically, GDEVs were found to inhibit IκBα degradation (NF-κB pathway), c-JUN N-terminal kinase (JNK) phosphorylation (MAPK pathway), and downstream kinase signaling cascades. Notably, the observed anti-osteoclastogenic effects may be attributed to bioactive constituents within GDEVs, particularly ginsenosides Rb1 and Rg1, which are known to possess potent immunomodulatory properties. Moreover, *Pueraria lobata*-derived EVs have been shown to enhance autophagy and stimulate the differentiation and function activity of human bone MSCs via gut microbiota-derived metabolites. These EVs exhibit dual therapeutic effects by promoting osteogenesis while suppressing bone resorption, thereby demonstrating significant potential in OP treatment [56]. These findings highlight PDEVs as a promising novel therapeutic strategy for OP. Chinese yam-derived EVs promote osteoblast proliferation, differentiation, and mineralization through activation of the BMP-2/p-p38-dependent Runx2 pathway [57]. Similarly, Cissus Quadrangularis-derived EVs were able to promote the proliferation and differentiation of MSCs and C2C12 cells into an osteoblastic cell lineage [58]. Furthermore, plum-derived EVs can enhance osteoblast differentiation and inhibit osteoclast activation [59], and EVs derived from apples were able to modulate the BMP2/Smad1 pathway, thereby enhancing osteogenic viability of MC3T3-E1 osteoblasts [60]. Finally, Rhizoma Drynariae-derived EVs exhibited potent bone tissue-targeting activity and anti-osteoporosis efficacy in an ovariectomized mouse model, and potentiated osteogenic differentiation of human bone MSCs by targeting estrogen receptor-alpha (ERα) [61]. PDEVs encapsulate natural bioactive components that enhance osteoblast proliferation and differentiation while suppressing osteoblast activity, thereby improving bone mineral density and microarchitecture. Mechanistically, PDEVs exert their therapeutic effects by modulating key signaling pathways involved in bone metabolism, highlighting their critical role in regulating physiological homeostasis. These findings position PDEVs as a promising therapeutic strategy for OP. Further research may establish PDEVs as a clinically viable treatment option for this debilitating condition.

##### Sarcopenia

Sarcopenia is an age-related syndrome characterized by progressive loss of skeletal muscle mass and strength. The pathogenesis involves multiple factors, including chronic physical inactivity, functional impairment, and nutritional deficiencies [62]. Notably, Gouqi-derived EVs have demonstrated therapeutic potential by increasing quadricep cross-sectional area and grip strength through activation of the AMPK/SIRT1/PGC1α signaling pathway. Furthermore, these EVs enhance amino sugar and nucleotide sugar metabolism while promoting autophagy and oxidative phosphorylation, collectively contributing to muscle regeneration [63]. Certain components in PDEVs, such as growth factors and cytokines, can promote the proliferation and differentiation of muscle cells, thereby helping to increase muscle mass. They can also inhibit muscle cell apoptosis by regulating the expression of apoptosis-related genes to reduce the loss of muscle tissue. Additionally, they can protect muscle cells from damage through anti-inflammatory and antioxidant stress mechanisms. Current research indicates that PDEVs show promising application prospects in the treatment of musculoskeletal system diseases, but their specific role and mechanisms in sarcopenia still require further in-depth study.

### 4.2. Application of PDEVs in Cardiovascular System Diseases

Cardiovascular diseases, a group of disorders affecting the circulatory system, have emerged as the leading global cause of mortality, surpassing cancer in both prevalence and fatality rates. This disease category encompasses various pathological conditions including atherosclerosis, myocardial infarction, cerebral ischemia, and heart failure, which collectively pose an escalating threat to global public health. Ongoing socioeconomic development and demographic aging have contributed to the increasing prevalence of modifiable risk factors such as tobacco use, unhealthy dietary patterns, and chronic stress, thereby exacerbating the cardiovascular disease burden [64]. The prevalence of cardiovascular diseases is still rising, urgently necessitating further research and development of drugs for cardiovascular treatment. EVs have been demonstrated to exert pleiotropic biological effects, including modulation of apoptosis, cellular proliferation, differentiation, and migration, as well as angiogenesis, anti-aging, and anti-inflammatory activities. These properties underscore their significant therapeutic potential in cardiovascular disease management. Specifically, EVs derived from endothelial progenitor cells, bone marrow macrophages, and thrombin-activated platelets have emerged as promising vehicles for targeted microRNA delivery in atherosclerosis therapy. However, the clinical application of EVs in atherosclerosis is challenging and hindered by challenges related to their source, production scale, acquisition complexity, and ethical considerations. In contrast, PDEVs offer distinct advantages, including broad source availability, high cost-effectiveness, superior production yield, excellent biocompatibility, and low immunogenicity. Structurally analogous to MDEVs, PDEVs represent a promising alternative and a complementary platform for advancing MDEV-based therapeutic development. *Carthamus tinctorius* L. has cardiovascular protective effects and is often used in the treatment of atherosclerosis-related diseases. Yang et al. [65] successfully isolated and characterized EVs derived from *Carthamus tinctorius* L., with particular focus on their miR166a-3p cargo. Their study systematically investigated the therapeutic potential of these PDEVs in ApoE−/− mice models of atherosclerosis induced by a high-fat diet. The researchers employed comprehensive experimental approaches, including the construction of miRNA mimics (agomiR166a-3p) and inhibitors (antagomiR166a-3p) for oral administration, to evaluate their effects on atherosclerotic progression. The study demonstrated that *Carthamus tinctorius* L.-derived EVs are effectively absorbed into systemic circulation through the gastrointestinal tract and display tissue-specific biodistribution patterns. Mechanistically, these EVs appear to attenuate inflammation in ox-LDL-treated HUVECs, potentially through modulation of the miR166a-3p/CXCL12 signaling axis, thereby conferring significant protection against miR166a-3p-mediated atherosclerosis pathogenesis. Avocado-derived EVs demonstrated significant anti-atherogenic effects by suppressing two key inflammatory pathways: (1) inhibition of NF-κB activation and NLRP3 inflammasome assembly, and (2) downregulation of pro-inflammatory and pro-atherogenic gene expression. Furthermore, these EVs effectively attenuated oxidized low-density lipoprotein (ox-LDL)-induced foam cell formation in macrophages. Mechanistically, while they did not interfere with ox-LDL binding to cell surface receptors, they specifically inhibited the subsequent internalization of ox-LDL during foam cell development. [66]. Oxidative stress is associated with many diseases, particularly cardiovascular and neurodegenerative diseases. Recent evidence demonstrates that carrot-derived EVs mediate intercellular communication through the transfer of bioactive molecules to recipient mammalian cells, subsequently modulating their biological activity. They can inhibit ROS production in rat embryonic cardiomyocytes and suppress apoptosis caused by oxidative stress. Additionally, they can effectively inhibit the reduction of antioxidant proteins to suppress apoptosis induced by oxidative stress [67]. This indicates the potential use of carrot-derived EVs in treating diseases related to abnormal apoptosis and ROS production, such as myocardial infarction. They could be used as a candidate drug for treating myocardial infarction. *Salvia miltiorrhiza*-derived EVs enhance the cell viability of human umbilical vein endothelial cells (HUVECs) and can promote cell migration and improve neovascularization in myocardial ischemia-reperfusion-injured cardiac tissues [68]. This indicates that they have great potential as angiogenesis enhancers in the treatment of myocardial ischemia-reperfusion injury. The therapeutic potential of PDEVs stems from their diverse cargo of bioactive molecules (including proteins, lipids, and nucleic acids) that modulate multiple cardiovascular processes: (1) regulation of cardiomyocyte apoptosis and survival pathways, (2) promotion of cardiac tissue repair and regeneration, (3) modulation of vascular tone through effects on diastolic and systolic functions, and (4) influence on vascular remodeling via regulation of permeability and neointima formation. However, the application of PDEVs in cardiovascular therapeutics remains in its preliminary stages. While preclinical results are encouraging, substantial further investigation is required, including (i) comprehensive mechanistic studies to elucidate molecular pathways, (ii) standardized protocols for EV isolation and characterization, and (iii) rigorous clinical trials to establish safety profiles and therapeutic efficacy in human patients, as depicted in Table 2 and Figure 6.

### 4.3. Application of PDEVs in Neurodegenerative Diseases

Neurodegenerative diseases, such as Alzheimer’s disease, Parkinson’s disease, Amyotrophic lateral sclerosis, and Huntington’s disease, are a heterogeneous group of disorders characterized by gradual progression and selective loss of anatomically or physiologically related neurons, significantly impairing cognitive or behavioral abilities and affecting millions of people worldwide [69]. Much effort has been invested in research related to these diseases, but few breakthroughs have been made in diagnostic and therapeutic approaches. In recent years, there has been a significant increase in research on the clinical applications of PDEVs, with their potential roles as diagnostic markers, therapeutic agents, and drug delivery vehicles receiving considerable attention. Their low immunogenicity and ability to cross the blood–brain barrier have also drawn increasing interest in their emerging roles in neurodegenerative diseases. Parkinson’s disease is a chronic neurological disorder, caused by an increase in reactive oxygen species during the degeneration of dopaminergic neurons. Carrot-derived EVs have significant antioxidant and apoptosis-inhibiting effects. Kim et al. [67] systematically evaluated the neuroprotective potential of carrot-derived EVs in a 6-hydroxydopamine (6-OHDA)-treated human neuroblastoma (SH-SY5Y) cell model. Their findings demonstrated that carrot-derived EVs exhibit minimal cytotoxicity in both H9C2 cardiomyocytes and SH-SY5Y neuronal cells at high concentrations, and significantly attenuate 6-OHDA-induced reactive oxygen species (ROS) generation and apoptosis in vitro. These results suggest that carrot-derived EVs may serve as promising therapeutic candidates for Parkinson’s disease intervention. Parallel research on EVs derived from the medicinal plant Salvia Hairy Roots was able to enter dopaminergic neurons, attenuate 6-OHDA-induced neurotoxicity, and prevent 6-OHDA-induced metabolomic changes. They can also prevent 6-OHDA autoxidation and pyrroloquinoline quinone formation and significantly inhibit apoptosis in a cell model of Parkinson’s disease, demonstrating the significant and multifaceted neuroprotective effects of EVs from Salvia Hairy Root. This suggests that EVs derived from Salvia Hairy Root could be used as a novel therapy for the treatment of neurological disorders [70].

Cerebral ischemia, a prevalent and life-threatening cardiovascular disorder, is characterized by high incidence and mortality rates. Currently, the primary clinical intervention for cerebral ischemia involves thrombolysis therapy, which facilitates the restoration of blood flow and oxygen supply to ischemic brain regions. However, abrupt recanalization of occluded arteries may induce excessive generation of reactive oxygen species (ROS), initiate inflammatory cascades, and subsequently exacerbate cellular damage in vulnerable tissues—a phenomenon termed cerebral ischemia-reperfusion injury. Notably, severe ischemic events and consequent brain damage can result in significant neurological impairments and functional deficits. *Panax notoginseng*-derived EVs attenuate cerebral ischemia-reperfusion injury by changing the phenotype of microglia from a “pro-inflammatory” M1 type to an “anti-inflammatory” M2 type. In addition, lipids are the main active ingredients in the treatment. It exerts its therapeutic effects by activating the PI3k/Akt pathway, improving infarct size, improving behavioral outcomes, and maintaining the integrity of the blood–brain barrier [71].

Fibrosis is a progressive degenerative disorder characterized by excessive extracellular matrix deposition. Kantarcıoğlu et al. [72] investigated the therapeutic potential of coffee-derived extracellular vesicles (EVs) by evaluating their early (24 h) and late (72 h) effects on two distinct cell lines: LX2 (hepatic stellate cells, a fibroblastic lineage) and HEP40 (hepatocellular carcinoma cells). Their findings revealed that coffee EVs significantly suppressed hepatocellular carcinoma cell proliferation in the early phase (24 h), whereas no significant inhibitory effect was observed at the later time point (72 h). They identified 15 novel miRNA sequences in coffee-derived EVs using the microRNA target prediction algorithm of the artificial intelligence program MapReduce for these novel miRNA sequences. They discovered two important genes, ZNF773 and KMT2C, and their network in liver fibrosis leading to chronic liver disease. Regular coffee consumption has beneficial and preventive effects on the liver and chronic neurodegenerative diseases. Coffee-derived EVs are new candidate drugs for the treatment of chronic liver disease, hepatocellular carcinoma, and chronic neurodegenerative diseases. Ginger-derived EVs have been shown to have neuroprotective effects. They can ameliorate neurodegenerative diseases by modulating inflammatory responses, resisting oxidative stress, and inhibiting neuronal degeneration. PDEVs can cross the blood–brain barrier and can serve as novel carriers for phytochemicals to deliver drugs for treating neurodegenerative diseases. This provides new ideas for addressing issues such as poor drug stability, low bioavailability, and insufficient drug accumulation in brain tissue. PDEVs do not contain any zoonotic or human pathogens, and they have a lower immunogenic risk in vivo, making them safer for the treatment of neurodegenerative diseases. Compared with MDEVs, PDEVs are rich in resources, low in production cost, and high in yield, which are conducive to large-scale production and application. They have enormous potential in the treatment of neurodegenerative diseases and are expected to provide new effective means for treating such diseases.

### 4.4. Application of PDEVs in Cancer

Cancer is a complex disease that is facilitated by many factors, including genetic and environmental factors, and ranks among the primary causes of morbidity and mortality worldwide. As the most important and effective existing methods for cancer treatment, radiotherapy and chemotherapy have played a great role in improving the survival rate and quality of life of cancer patients, but they also show great toxicity [73]. PDEVs specifically not only inhibit or kill rapidly proliferating cancer cells, but also act on rapidly proliferating normal cells, such as bone marrow, hair follicle, and gastrointestinal tract cells. Therefore, it is imperative to explore new therapeutic approaches to achieve effective anti-cancer results. PDEVs have emerged as a promising avenue in the realm of oncological research, owing to the combination of their natural derivation, outstanding biocompatibility, innate tumor-targeting faculties, and adeptness in precision delivery of therapeutic agents. For instance, PDEVs derived from cucumbers possess bioactive secondary metabolites, notably cucurbitonin B, which has been documented to impede the progression of leukemia, breast cancer, lung cancer, and liver cancer [74]. Furthermore, the presence of sulforaphane compounds in broccoli-derived PDEVs has been found to hinder the development of diverse cancers, such as pancreatic, intestinal, leukemia, and prostate cancers [75]. Han et al. [76] demonstrated that ginseng-derived EVs possess immunomodulatory properties capable of delaying the progression of B16F10 melanoma under heat-induced conditions. These nanovesicles were found to reprogram tumor-associated macrophages by shifting their polarization from the pro-tumor M2 phenotype, thereby enhancing the secretion of chemokines CCL5 and CXCL9. This chemokine upregulation promoted the recruitment of CD8+ T cells to the tumor site, ultimately reshaping the tumor microenvironment and exerting potent anti-tumor effects. Moreover, ginger-derived EVs have been shown to reduce IL-6 and IL-1β levels in colorectal cancer mouse models, as well as to inhibit the mRNA expression of TNF-α and cyclin D1, thereby suppressing tumor cell proliferation [77]. From this, it can be seen that PDEVs demonstrate remarkable potential in cancer therapy, exhibiting not only broad-spectrum anti-tumor effects but also favorable biosafety profiles characterized by low immunogenicity, minimal toxicity, and negligible side effects.
biomolecules-15-01095-t002_Table 2Table 2Applications and mechanisms of PDEVs in regenerative medicine.DiseaseSource of PDEVsTreating DiseasesMechanisms of TreatmentReferencesTissue repair and wound healingSaponariaTissue repairReduced expression of pro-inflammatory genes promotes blood vessel formation in HUVECs[46]
*Momordica charantia*Tissue repairPromotes cell proliferation and inhibits apoptosis[47]
GingerTissue repairStimulation of the Wnt/β-catenin signaling pathway leads to an increased population of intestinal epithelial stem cells[48]
*Beta vulgaris*Tissue repairPromotes vasculature-like network and inhibits fibroblast migration[78]
*Opuntia ficus-indica* FruitHealing of a woundReduced pro-inflammatory cytokine activity and gene expression[79]
GinsengHealing of a woundStimulate cell proliferation and migration, express wound healing-related genes, and promote angiogenesis[80]
*Pueraria lobata*Healing of a woundPromote the polarization of M2 macrophages to exert their anti-inflammatory effect[56]
Loquat LeavesTissue repairInhibit apoptosis of human skin fibroblasts; restore cell migration ability[81]
LemonTissue repairReduce fibroblast ROS[82]
WheatHealing of a woundIncrease cell migration; reduce cell apoptosis[83]
*Cissus quadrangularis*Healing of a woundIncrease the migration ability of natural cells[58]Musculoskeletal system diseasesSpinachOsteoarthritisIncreases intracellular ATP and NADPH levels under natural light and improves anabolism in degraded chondrocytes[51]
TomatoOsteoarthritisIncreased expression of key soft markers (ACAN, SOX9, and COMP) and key proteins (COL2 and COLXI) in chondrocytes[52]
Curcumin reinforces mesenchymal stem cells-derived EVsOsteoarthritisSignificant upregulation of miR-143 and miR-124 expression and downregulation of NF-kB and ROCK1 expression in osteoarthritis cells[53]
GinsengOsteoporosisInhibit the differentiation of osteoclasts[45]
*Pueraria lobata*OsteoporosisDifferentiation of mesenchymal stem cells and C2C12 cells into the osteogenic lineage[56]
Chinese YamOsteoporosisPromote the proliferation, differentiation, and mineralization of osteoblasts[57]
*Cissus quadrangularis*OsteoporosisPromote the proliferation and differentiation of mesenchymal stem cells and C2C12 cells into the osteogenic lineage[58]
PlumOsteoporosisImprove osteoblast differentiation and inhibit osteoclast activation[59]
AppleOsteoporosisEnhance the osteogenic ability of MC3T3-E1 osteoblasts[60]
Rhizoma DrynariaeOsteoporosisPotentiated osteogenic differentiation of hBMSCs by targeting ERα[61]
GouqiSarcopeniaImproved cross-sectional area, grip strength, and AMPK/SIRT1/PGC1α pathway expression[63]Diseases of the cardiovascular system*Carthamus tinctorius* L.AtherosclerosisAnti-inflammatory effects on ox-LDL-treated HUVECs via the miR166a-3p/CXCL12 pathway[65]
*Pueraria lobata*AtherosclerosisInhibition of NF-κB and NLRP3 activation and suppression of pro-inflammatory and atherogenic gene expression[56]
CarrotMyocardial infarctionInhibition of ROS production and suppression of oxidative stress-induced apoptosis in rat embryonic cardiomyocytes[67]
*Salvia miltiorrhiza*Myocardial ischemiaEnhancement of human umbilical vein endothelial cell viability and cell migration[68]Neurodegenerative diseaseCarrotParkinson’s diseaseInhibition of ROS production and apoptosis in vitro[67]
Salvia Hairy RootParkinson’s diseaseInhibition of 6-hydroxydopamine autoxidation and pyrroloquinoline quinone formation and apoptosis[70]
*Panax notoginseng*Cerebral infarctionActivation of the pI3k/Akt pathway exerts therapeutic effects[71]
CoffeeLiver fibrosisInhibition of hepatocellular carcinoma cell proliferation[72]CancerGinsengMelanomaDelaying the progression of B16F10 melanoma under heat-induced conditions[76]
GingerColorectal cancerInhibit the mRNA expression of TNF-α and cyclin D1[77]
Tea flowerBreast cancerInhibited the tumor growth in breast cancer[84]
*Morus nigra* L.Liver cancerInduced apoptosis in Hepa1–6 cells[85]

## 5. Clinical Trials of PDEVs

Although people’s enthusiasm is growing day by day, currently only a few clinical trials have begun to explore the characteristics of PDEVs. The first clinical attempt was started in August 2012, and was focused on evaluating the effectiveness of grape-derived EVs in preventing oral mucositis in patients undergoing chemoradiation treatment for head and neck cancer [86]. The second item is a study that focuses on investigating the ability of PDEVs to deliver curcumin to normal and colon cancer tissue [87]. However, as the research is still in its infancy, there are very few clinical trials on PDEVs in progress or completed; see Table 3 for details. PDEVs, as novel therapeutic materials, have demonstrated significant potential in laboratory studies. However, their transition from the lab to clinical and commercial applications still faces multiple technological and regulatory challenges. First, in terms of technology, the large-scale production and standardization of PDEVs present numerous issues, such as relatively low yields that are insufficient to meet clinical demand, difficulties in controlling extraction heterogeneity, and contamination. Second, PDEVs exhibit low drug-loading efficiency and lack active targeting capabilities. Finally, challenges remain regarding the stability and storage methods of PDEVs. For instance, freeze-dried PDEVs tend to aggregate, and their membrane structure is prone to degradation at room temperature.

## 6. Conclusions and Outlook

PDEVs exert multifaceted therapeutic effects on skeletal muscle through several distinct mechanisms. First, the antioxidant and anti-inflammatory components contained in PDEVs effectively inhibit pro-inflammatory cytokine secretion, attenuate oxidative stress-induced skeletal muscle damage, and mitigate neuronal degeneration, thereby ameliorating neurodegenerative pathologies. Second, PDEVs demonstrate remarkable capacity to promote tissue regeneration by stimulating the proliferation and differentiation of various cell lineages, including osteoblasts, fibroblasts, and chondrocytes, while also facilitating cellular polarization. These processes collectively contribute to the repair and regeneration of damaged skeletal muscle tissue. Furthermore, PDEVs enhance vascularization in skeletal muscle by increasing capillary branching and vascular density, thereby improving local blood perfusion and tissue oxygenation. Finally, the immunomodulatory factors present in PDEVs play a crucial role in regulating immune cell activity and function. By suppressing aberrant autoimmune responses, PDEVs protect skeletal muscle from immune-mediated damage and help maintain its physiological integrity. This comprehensive mechanism underscores the therapeutic potential of PDEVs in skeletal muscle repair and regeneration.

Conventional regenerative medicine therapies face several critical limitations that hinder their clinical efficacy. First, many traditional pharmacological agents induce significant adverse effects during treatment, compromising both patient quality of life and therapeutic outcomes. Furthermore, prolonged administration of certain drugs may lead to the development of drug tolerance, resulting in diminished efficacy or complete therapeutic failure over time. Current drug delivery systems also present substantial challenges. Oral formulations often suffer from poor bioavailability, while injectable alternatives cause patient discomfort and inconvenience. Additionally, many therapeutic compounds cannot effectively traverse biological barriers—such as the blood–brain barrier or cutaneous barrier—severely limiting their ability to reach target tissues. Beyond pharmacological limitations, existing treatment modalities encounter major obstacles, including technical difficulties in stem cell culture, complications arising from excessive therapeutic intervention, prohibitive costs, and unpredictable treatment outcomes. These constraints underscore the urgent need for more advanced and reliable therapeutic strategies in regenerative medicine. PDEVs offer distinct advantages as therapeutic agents due to their natural origin and sustainable production potential. As naturally occurring nanocarriers, PDEVs demonstrate superior biocompatibility and low immunogenicity in biological systems. Their intrinsic targeting capability enables cell-specific delivery of bioactive cargo, facilitating precise therapeutic intervention while minimizing off-target effects. This targeted delivery mechanism significantly reduces the risk of immune reactions and graft rejection compared to conventional therapies. Consequently, PDEV-based treatments are associated with fewer procedural risks and complications, ultimately leading to enhanced therapeutic efficacy. The combination of these properties positions PDEVs as a promising platform for next-generation drug delivery systems in regenerative medicine. PDEVs exhibit remarkable therapeutic potential through multiple mechanisms. First, they actively promote cellular processes essential for tissue regeneration, including (1) enhanced proliferation and migration of progenitor cells, (2) directed differentiation into multiple tissue lineages, and (3) modulation of immune responses through antigen presentation. PDEVs achieve immunomodulation through a dual mechanism: by carrying antigenic molecules to stimulate immune cells to generate immune responses. Through immunosuppressive regulation, they create a favorable microenvironment for tissue repair. They hold great potential in tissue repair and regeneration. PDEVs can cross cellular barriers, such as the blood–brain and skin barriers, to deliver drugs or bioactive molecules to target tissues or organs. This ability to cross biological barriers enables them to more effectively deliver drugs to diseased areas, thereby enhancing the bioavailability and therapeutic effect of the drugs. Compared with MDEVs, PDEVs have significant inherent advantages, while MDEVs face many limitations. First of all, because MDEV-enriched fluids usually contain multiple impurities and have similar sizes and densities, which lead to difficulties in characterization, better methods need to be developed to quantify and characterize MDEVs. Secondly, the low yield of EVs is also a challenge. Furthermore, the utilization rate of MDEVs is low, they are easily degraded, and it is difficult to maintain their function for a long period; therefore, it is necessary to use biomaterials loaded with MDEVs to slow down their degradation [88]. Finally, MDEVs also face immunogenic risks, such as bacterial-derived EVs potentially containing lipopolysaccharides or other pathogen-associated molecules, which can easily trigger a strong immune response or sepsis risk. On the contrary, the extraction and preparation costs of PDEVs are lower and easier to scale up for production. As renewable resources, their extraction and benefits do not have a serious impact on the environment, aligning with the concept of sustainable development. PDEVs have multiple advantages in regenerative medicine, such as natural origin and biocompatibility, diverse bioactive components, potential as drug carriers, ability to promote tissue repair and regeneration, ability to cross biological barriers, as well as economic benefits and sustainability. PDEVs have a broad application prospect and great development potential in regenerative medicine. PDEVs exhibit distinct advantages in clinical translation, particularly in terms of biocompatibility, safety profile, and cross-species applicability. (1) Their inherent biocompatibility and low toxicity stem from significant differences between plant-derived surface proteins and human proteins, which not only minimize immune rejection reactions but also enable repeated administration. This characteristic, combined with the absence of animal virus contamination, ensures superior safety. (2) PDEVs serve as remarkably stable drug delivery platforms, capable of encapsulating diverse therapeutic agents including hydrophilic/hydrophobic drugs, nucleic acids, and natural bioactive molecules. They demonstrate exceptional stability in simulated gastrointestinal environments and maintain integrity through freeze–thaw cycles [77]. (3) Rich in natural bioactive components, PDEVs possess intrinsic targeting capabilities. For instance, ginger-derived EVs loaded with siRNA and chemotherapeutic agents have demonstrated precise delivery to target cancer cells while sparing major organs, concurrently mitigating conventional chemotherapy-induced adverse effects [89]. (4) The scalability of PDEV production is facilitated by their abundant sources and cost-effectiveness, making them particularly suitable for large-scale clinical applications.

Despite their therapeutic potential, research on PDEVs remains in its nascent stage, with several critical challenges requiring resolution: 1. Key Research Gaps: (1) Absence of standardized biomarkers and characterization protocols. (2) There remains a limited understanding of their biodistribution and pharmacokinetic profiles, particularly regarding carrier–drug interactions. Key challenges include inefficient targeting, limited ability to cross biological barriers, difficulty in achieving an optimal balance between controlled release and systemic clearance, and complex interactions between the carrier and the drug. (3) Limited mechanistic studies on their therapeutic actions. (4) Lack of comprehensive safety evaluation and regulatory frameworks. (5) Technical challenges in scalable isolation and purification methods. 2. Critical Development Needs: (1) Establishment of Good Manufacturing Practice (GMP) standards. (2) Development of quantitative analytical methods. (3) Implementation of quality control parameters. (4) Creation of standardized storage and stability protocols. 3. Future Research Priorities: (1) Elucidation of structure–function relationships. (2) Investigation of long-term biological effects. (3) Optimization of cargo loading efficiency. (4) Development of targeted modification strategies. As fundamental research progresses, PDEVs show tremendous promise for revolutionizing regenerative medicine by (1) providing novel therapeutic modalities for tissue repair, (2) enabling targeted drug delivery across biological barriers, and (3) offering cost-effective alternatives to current cell-based therapies. Currently, PDEVs exhibit both promise and controversy. Key issues include the following: (1) Nomenclature and Classification Ambiguity: The terminology for PDEVs remains inconsistent, often overlapping with terms such as “plant nanovesicles (PNVs)” or “plant exosome-like nanoparticles (PELNs).” (2) Lack of Universal Markers and Compositional Heterogeneity: Unlike mammalian exosomes, PDEVs lack well-defined surface markers (e.g., CD9, CD63), and their lipid, protein, and nucleic acid profiles vary significantly across plant species. (3) Unclear Cross-Kingdom Regulatory Mechanisms: Although PDEVs can mediate cross-kingdom gene regulation, their precise molecular targets and signaling pathways require further validation. Additionally, challenges in safety and clinical translation persist, including the following: Uncertainty in Administration Routes and Pharmacokinetic Stability: The oral bioavailability, gastrointestinal stability, and targeting efficiency of PDEVs remain poorly understood. Limited Clinical Evidence: Most studies are confined to preclinical models, with only a few PDEVs advancing to clinical trials. Addressing these issues is critical for advancing PDEVs toward therapeutic applications. The resolution of these challenges will pave the way for PDEVs to emerge as a transformative platform in precision medicine, potentially addressing unmet needs in various therapeutic areas while maintaining sustainable production standards. Continued investment in PDEV research promises to yield significant advancements in regenerative therapies within the coming decade.

## Figures and Tables

**Figure 1 biomolecules-15-01095-f001:**
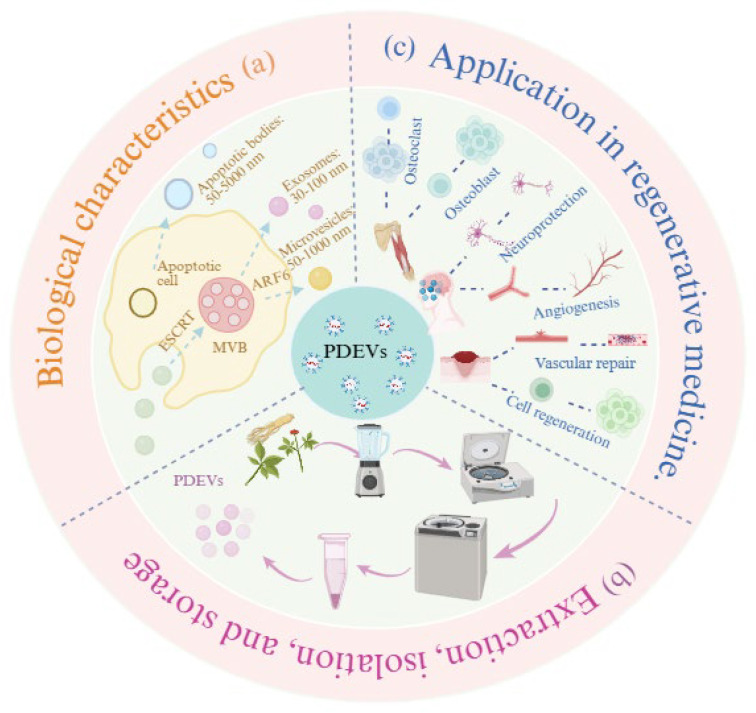
The biological characteristics, extraction methods, and applications of PDEVs in regenerative medicine. (**a**) Biological characterization of PDEVs. (**b**) Extraction, isolation, and storage of PDEVs. (**c**) Application of PDEVs in regenerative medicine.

**Figure 2 biomolecules-15-01095-f002:**
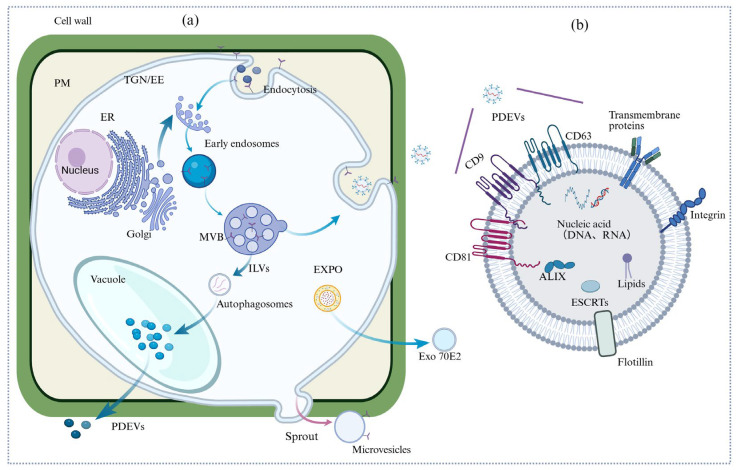
PDEV secretion mechanism diagram. (**a**) Biogenesis of PDEVs. (**b**) Structure of PDEVs.

**Figure 3 biomolecules-15-01095-f003:**
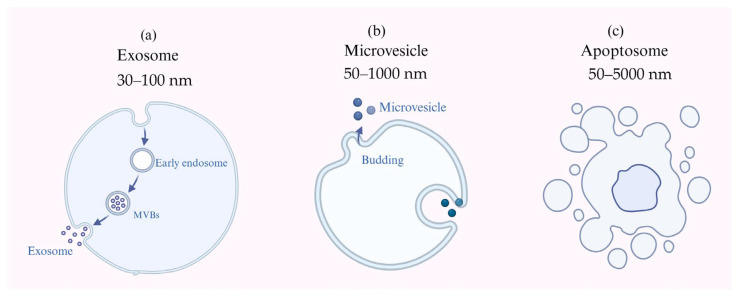
PDEV classification diagram. (**a**) Exosomes and their biogenesis. (**b**) Microvesicles and their biogenesis. (**c**) Apoptosomes and their biogenesis.

**Figure 4 biomolecules-15-01095-f004:**
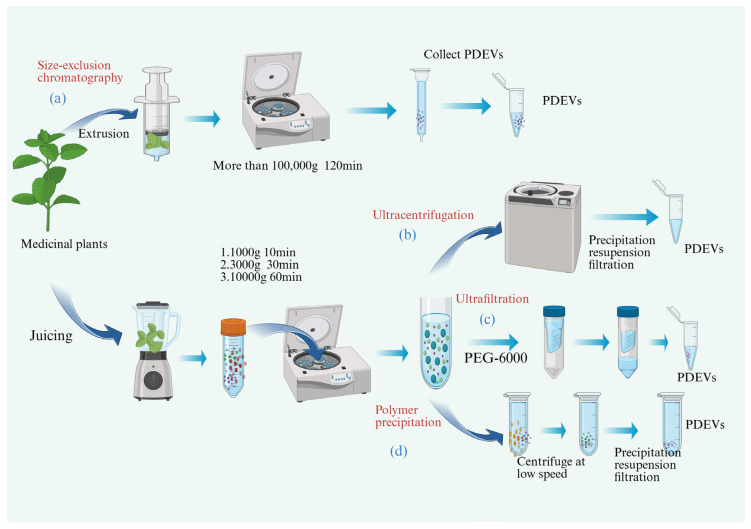
Diagram of commonly used extraction methods for PDEVs. (**a**) Size-exclusion chromatography for the extraction of PDEVs. (**b**) Ultracentrifugation method for extracting PDEVs. (**c**) Ultrafiltration method for extracting PDEVs. (**d**) Polymer precipitation method for extracting PDEVs.

**Figure 5 biomolecules-15-01095-f005:**
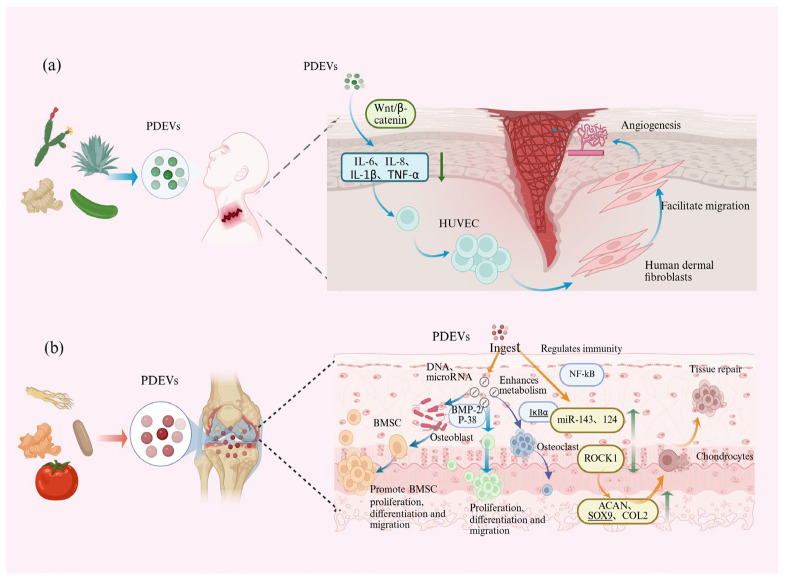
Schematic illustration of the therapeutic mechanisms of PDEVs in promoting wound healing and treating musculoskeletal system diseases. (**a**) Schematic illustration of the therapeutic mechanisms of PDEVs in promoting wound healing. (**b**) Schematic illustration of the therapeutic mechanisms of PDEVs in treating musculoskeletal system diseases. (↑ Indicate an increase, ↓ Indicating reduction).

**Figure 6 biomolecules-15-01095-f006:**
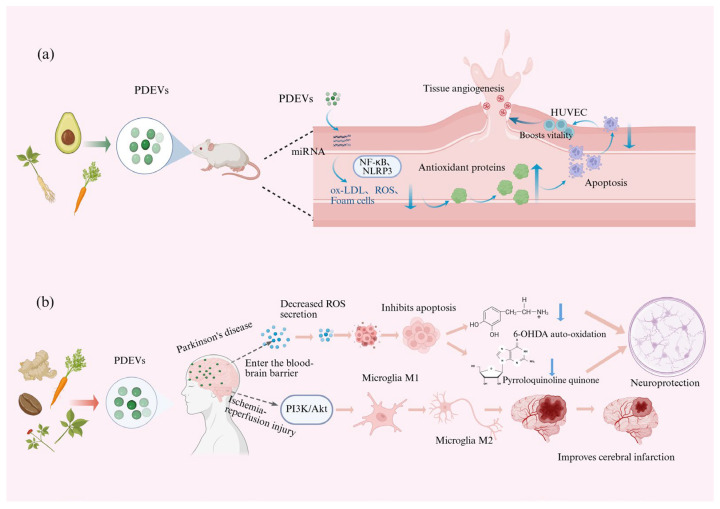
Mechanism diagram of PDEVs in the treatment of cardiovascular system diseases and neurodegenerative diseases. (**a**) Mechanism diagram of PDEVs in the treatment of musculoskeletal system cardiovascular system diseases. (**b**) Mechanism diagram of PDEVs in the treatment of neurodegenerative diseases. (↑ Indicate an increase, ↓ Indicating reduction).

**Table 3 biomolecules-15-01095-t003:** Two Clinical Trials Related to PDEVs.

PDEVs	Disease	Clinical Phase	Outcome	References
Grape-derived EVs	Head and neck cancer	Phase I	Effectiveness of grape-derived EVs in reducing oral mucositis in patients undergoing chemoradiotherapy for head and neck cancer.	[86]
Curcumin	Colon cancer tissue	Phase I	Delivery of Curcumin to Normal and Colon Cancer Tissues by PDEVs.	[87]

## Data Availability

Not applicable.

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
