# Peer review of "A New Perspective on Regenerative Medicine: Plant-Derived Extracellular Vesicles"

_biomolecules, 2025, doi:10.3390/biom15081095_

Round 1

Reviewer 1 Report

Comments and Suggestions for Authors

In this review, the authors present a comprehensive overview of PDEVs, systematically examining their biological properties, standardized methodologies for isolation and characterization, preservation techniques, and current applications in regenerative medicine. Furthermore, the review discusses emerging research directions and the clinical translation potential of PDEVs, with the aim of advancing their development as novel therapeutic agents. A major revision is recommended.

Specific comments are as follows:

  1. The authors should provide a comparative analysis between PDEVs and EVs derived from other sources—such as bacterial and mammalian cells—to highlight the unique features, advantages, and limitations of PDEVs.
  2. The section on purification, isolation, or storage should be expanded to explicitly discuss how the challenges associated with PDEVs differ from those encountered with EVs from other sources.
  3. The scope of therapeutic applications should be broadened to include cancer treatment.
  4. Applications in skeletal disorders, such as cartilage and bone repair, should be categorized under the broader theme of tissue regeneration.
  5. The authors are encouraged to include tables for the current clinical trial of PDEVs with detailed information.
  6. The authors are encouraged to refer to more relevant literature on EV clinical applications (e.g; 1) Engineering therapeutical extracellular vesicles for clinical translation 10.1016/j.tibtech.2024.08.007; ii) Extracellular Vesicles as Therapeutic Resources in the Clinical Environment doi.org/10.3390/ijms24032344) to expand this topic. Be specific about the challenges—both technological and regulatory—that impede the bench-to-market transition for PDEV-based therapies.
  7. Any special challenges or unique translation advantages for PDEVs? Please be specific.

Author Response

Comments 1 : The authors should provide a comparative analysis between PDEVs and EVs derived from other sources—such as bacterial and mammalian cells—to highlight the unique features, advantages, and limitations of PDEVs.

Response 1 : Thank you very much for the experts' suggestions. We have made revisions and additions to the content in the 6. Conclusion and outlook section. In the second paragraph of this section, the authors have added the limitations faced by MDEVs and compared them with PDEVs, further highlighting the advantages of PDEVs. The revised content is as follows: Compared with MDEVs, PDEVs have significant inherent advantages, while MDEVs face many limitations. First of all, because MDEVs-enriched fluids usually contain multiple impurities and have similar sizes and densities, which lead to difficulties in characterization, better methods need to be developed to better quantify and characterize MDEVs. Secondly, the low yield of EVs is also a challenge. Furthermore, the utilization rate of MDEVs is low, the are easy to be degraded and difficult to function for a long period, therefore, it is necessary to use biomaterials loaded with MDEVs to slow down their [91]. Finally, MDEVs also face immunogenic risks, such as bacterial-derived EVs potentially containing lipopolysaccharides or other pathogen-associated molecules, which can easily trigger a strong immune response or sepsis risk.

Comments 2: The section on purification, isolation, or storage should be expanded to explicitly discuss how the challenges associated with PDEVs differ from those encountered with EVs from other sources.

Response 2 : Thank you very much for the experts' suggestions. We have made revisions to the 3. Extraction, isolation, and storage of PDEVs section. The modifications include the challenges faced by MDEVs during extraction, further highlighting the advantages of PDEVs. Additionally, we have incorporated the strengths and weaknesses of commonly used extraction methods for PDEVs, making this section more comprehensive and further emphasizing the benefits of PDEVs. The revised content is as follows: PDEVs have similar functions to MDEVs, but due to the complexity of the biological fluid, contamination often occurs with non-vesicular components such as lipoprotein and nucleoprotein complexes [28]. That lead the purification steps for MDEVs pose obstacles to separating them from the supernatant of tissue cultures. Therefore, PDEVs are becoming more attractive because they can overcome these defects. (1) The extraction of PDEVs is simpler and more cost-effective compared to MDEVs, as it does not rely on complex cell culture systems. Moreover, (2) PDEVs can be readily obtained from abundant plant sources, significantly reducing production costs. (3) Another key advantage is their lower risk of pathogen contamination, enhancing their safety profile. (4) Furthermore, PDEVs exhibit strong potential for scalable manufacturing, making them a promising candidate for industrial and biomedical applications. PDEVs can be extracted from plant tissues through mechanical disruption methods such as juicing or extrusion. Currently, multiple isolation techniques are employed to purify PDEVs, including ultracentrifugation, density gradient centrifugation, ultrafiltration, size exclusion chromatography, polymer precipitation, tangential flow filtration, and asymmetric flow field-flow fractionation. Furthermore, emerging technologies such as microfluidic systems, nano-flow cytometry, and nanoscale lateral displacement array are increasingly being utilized for high-precision isolation and characterization of PDEVs. Ultracentrifugation is simple to perform and inexpensive since it does not require solvents or substances. However, this process is too time-consuming and requires expensive machinery [29]. ultrafiltration is a size-based technique involving membrane filters and pressure, resulting in faster separation and does not necessitate special equipment. The efficiency of this method is greater than that of ultracentrifugation, and the time required is shorter [30]. Size exclusion chromatography is a simple and economical method. Working with gravity does not affect the integrity of the PDEVs and results in great reproducibility; however, specific equipment is needed, its scalability is difficult, and it takes a long time to perform [31]. Polymer precipitation is easy to perform, cost-effective and does not affect the quality of the exosomes. However, due to vesicle aggregation, the process could result in co-isolation with other PDEVs or other cellular components [32].

As a novel class of therapeutic agents or delivery vehicles, PDEVs require optimized storage conditions to preserve their structural integrity and biological functionality. Studies indicate that PDEVs remain stable for approximately one year when stored at –80°C, whereas storage at –20°C maintains stability for up to three months [33]. Leng Y et al. [34] found that the optimal storage temperatures for blueberry-derived EVs were 4°C for short-term storage and -80°C for long-term storage. Storage at 4°C helped to prevent ice crystals from damaging the phospholipid bilayer membranes of the PDEVs, while storage at -80°C slowed down the rate of degradation and maintained the particle morphology. Richter M et al. [35] found that PDEVs stored at -80°C and 4°C had higher particle recovery than freeze-dried PDEVs.

Comments 3 : The scope of therapeutic applications should be broadened to include cancer treatment.

Response 3 : Thank you very much for the experts' valuable suggestions. The authors have added Section 4.4 Application of PDEVs in Cancer and made corresponding revisions to Table 2. The modified content is as follows: Cancer is a complex disease, that is, facilitated by many factors, including genetic and environmental factors, and ranking among the primary causes of morbidity and mortality worldwide. As the most important and effective methods for cancer treatment before, radiotherapy and chemotherapy have played a great role in improving the survival rate and quality of life of cancer patients, but they also show great toxicity [75]. PDEVs specifically, they not only inhibit or kill rapidly proliferating cancer cells, but also act on rapidly proliferating normal cells, such as bone marrow, hair follicle and gastrointestinal tract cells. Therefore, it is imperative to explore new therapeutic approaches to achieve effective anti-cancer results. PDEVs have emerged as a promising avenue in the realm of oncological research, owing to the combination of their natural derivation, outstanding biocompatibility, innate tumor-targeting faculties, and adeptness in precision delivery of therapeutic agents. For instance, PDEVs derived from cucumbers possess bioactive secondary metabolites, notably cucurbitonin B, which has been documented to impede the progression of leukemia, breast cancer, lung cancer, and liver cancer [76]. Furthermore, the presence of sulforaphane compounds in broccoli-derived PDEVs has been found to hinder the development of diverse cancers, such as pancreatic, intestinal, leukemia, and prostate cancers [77]. Han et al. [78] demonstrated that ginseng-derived EVs possess immunomodulatory properties capable of delaying the progression of B16F10 melanoma under heat-induced conditions. These nanovesicles were found to reprogram tumor-associated macrophages by shifting their polarization from the pro-tumor M2 phenotype, thereby enhancing the secretion of chemokines CCL5 and CXCL9. This chemokine upregulation promoted the recruitment of CD8+ T cells to the tumor site, ultimately reshaping the tumor microenvironment and exerting potent anti-tumor effects. Moreover, ginger-derived EVs have been shown to reduce IL-6 and IL-1β levels in colorectal cancer mouse models, as well as inhibit the mRNA expression of TNF-α and cyclin D1, thereby suppressing tumor cell proliferation [79] From this, it can be seen that PDEVs demonstrate remarkable potential in cancer therapy, exhibiting not only broad-spectrum anti-tumor effects but also favorable biosafety profiles characterized by low immunogenicity, minimal toxicity, and negligible side effects.

Comments 4 : Applications in skeletal disorders, such as cartilage and bone repair, should be categorized under the broader theme of tissue regeneration.

Response 4 : Thank you for the experts' suggestions. The section has been reorganized accordingly. In the manuscript, the section on the application of PDEVs in “skeletal muscle system disorders” has already been categorized under the “tissue regeneration” section.

Comments 5 : The authors are encouraged to include tables for the current clinical trial of PDEVs with detailed information.

Response 5 : Thank you for the experts' valuable suggestions. The authors have added a new section titled "5. Clinical Trials of PDEVs" in the manuscript. The specific modifications are as follows: Although people's enthusiasm is growing day by day, currently only a few clinical trials have begun to explore the characteristics of PDEVs. The first clinical attempt was started in August 2012, and was focused on evaluating the effectiveness of grape-derived EVs in preventing oral mucositis in patients undergoing chemoradiation treatment for head and neck cancer[92]. The second item is a study that focuses on investigating the ability of PDEVs to deliver curcumin to normal and colon cancer tissue. [93] However, as the research is still in its infancy, there are very few clinical trials on PDEVs in progress or completed, See Table 3 for details.

Table 3 Two Clinical Trials Related to PDEVs

PDEVs

Disease

Clinical Phase

Outcome

Literature

Grape-derived EVs

Head and neck cancer

Phase I

Effectiveness of grape-derived EVs in reducing oral mucositis in patients undergoing chemoradiotherapy for head and neck cancer.

[92]

curcumin

Colon cancer tissue

Phase I

Delivery of Curcumin to Normal and Colon Cancer Tissues by PDEVs.

[93]

Comments 6 : The authors are encouraged to refer to more relevant literature on EV clinical applications (e.g; 1) Engineering therapeutical extracellular vesicles for clinical translation 10.1016/j.tibtech.2024.08.007; ii) Extracellular Vesicles as Therapeutic Resources in the Clinical Environment doi.org/10.3390/ijms24032344) to expand this topic. Be specific about the challenges—both technological and regulatory—that impede the bench-to-market transition for PDEV-based therapies.

Response 6 : We sincerely appreciate the experts’ constructive comments. In response, the authors have incorporated relevant literature on the clinical applications of EVs into the manuscript. Additionally, Section 5: Clinical Trials of PDEVs has been expanded to discuss the challenges associated with translating PDEVs from laboratory research to clinical commercialization. The revised content is as follows: Although people's enthusiasm is growing day by day, currently only a few clinical trials have begun to explore the characteristics of PDEVs. The first clinical attempt was started in August 2012, and was focused on evaluating the effectiveness of grape-derived EVs in preventing oral mucositis in patients undergoing chemoradiation treatment for head and neck cancer [89]. The second item is a study that focuses on investigating the ability of PDEVs to deliver curcumin to normal and colon cancer tissue. [90] However, as the research is still in its infancy, there are very few clinical trials on PDEVs in progress or completed. PDEVs, as novel therapeutic materials, have demonstrated significant potential in laboratory studies. However, their transition from the lab to clinical and commercial applications still faces multiple technological and regulatory challenges. First, in terms of technology, the large-scale production and standardization of PDEVs present numerous issues, such as relatively low yields that are insufficient to meet clinical demand, difficulties in controlling extraction heterogeneity, and contamination. Second, PDEVs exhibit low drug-loading efficiency and lack active targeting capabilities. Finally, challenges remain regarding the stability and storage methods of PDEVs. For instance, freeze-dried PDEVs tend to aggregate, and their membrane structure is prone to degradation at room temperature.

Comments 7 : Any special challenges or unique translation advantages for PDEVs? Please be specific.

Response 7 : We sincerely appreciate the experts' insightful comments. In response to these valuable suggestions, the authors have enhanced Section 6: Conclusion and Outlook by incorporating a comprehensive discussion on both the advantages of PDEVs in clinical translation and the existing challenges. The specific modifications are as follows: PDEVs exhibit distinct advantages in clinical translation, particularly in terms of biocompatibility, safety profile, and cross-species applicability. (1) Their inherent biocompatibility and low toxicity stem from significant differences between plant-derived surface proteins and human proteins, which not only minimize immune rejection reactions but also enable repeated administration. This characteristic, combined with the absence of animal virus contamination, ensures superior safety. (2) PDEVs serve as remarkably stable drug delivery platforms, capable of encapsulating diverse therapeutic agents including hydrophilic/hydrophobic drugs, nucleic acids, and natural bioactive molecules. They demonstrate exceptional stability in simulated gastrointestinal environments and maintain integrity through freeze-thaw cycles [92]. (3) Rich in natural bioactive components, PDEVs possess intrinsic targeting capabilities. For instance, ginger-derived EVs loaded with siRNA and chemotherapeutic agents have demonstrated precise delivery to target cancer cells while sparing major organs, concurrently mitigating conventional chemotherapy-induced adverse effects [93]. (4) The scalability of PDEV production is facilitated by their abundant sources and cost-effectiveness, making them particularly suitable for large-scale clinical applications.

Despite their therapeutic potential, research on PDEVs remains in its nascent stage, with several critical challenges requiring resolution: 1. Key Research Gaps: (1) Absence of standardized biomarkers and characterization protocols. (2) Insufficient understanding of their biodistribution and pharmacokinetic profiles. (3) Limited mechanistic studies on their therapeutic actions. (4) Lack of comprehensive safety evaluation and regulatory frameworks. (5) Technical challenges in scalable isolation and purification methods.

Reviewer 2 Report

Comments and Suggestions for Authors

The manuscript attempts to provide a comprehensive review of plant-derived extracellular vesicles (PDEVs) and potential therapeutic applications in regenerative medicine. The topic is timely and relevant; however, the manuscript reads more like an encyclopedic article than a critical scientific review. Several areas require refinement to meet the standards expected for a peer-reviewed journal.

  1. The numbering and the structure are confusing to the reader. There is no section 2; however, subsections under section 3 are numbered as sections 2.1 and 2.2. Subsections under section 5 are numbered as 4.1.1. 4.1.2 ...
  2. There are plenty of grammatical and syntax errors all over the manuscript[pt that add difficulty to comprehension. 
  3. A lot of statements and claims are missing bibliographic documentation or  the references are not primary sources of information.
  4. Expressions like "great potential"  and "low immunogenicity" are used throughout the manuscript,  without explanation or specific documentation.
  5. Almost all citations originate from a narrow geographic region. I would suggest that you include landmark international studies and ISEV guidelines.
  6. The review does not adequately discuss controversies, limitations, or conflicting findings in the field.
  7. The authors  mention that PDEVs have similar or superior therapeutic potential compared to MDEVs, but without adequate supporting evidence
  8. There is no discussion of pharmacokinetics, biodistribution, GMP manufacturing, regulatory challenges, or ongoing clinical trials.

Author Response

Comments 1 : The numbering and the structure are confusing to the reader. There is no section 2; however, subsections under section 3 are numbered as sections 2.1 and 2.2. Subsections under section 5 are numbered as 4.1.1. 4.1.2 ...

Response 1 : Thank you for the valuable insights and concerns raised by the experts. I have now conducted a comprehensive revision of the entire manuscript structure.

Comments 2 : There are plenty of grammatical and syntax errors all over the manuscript[pt that add difficulty to comprehension. 

Response 2 : We sincerely appreciate the experts' valuable feedback. The manuscript has been meticulously refined by two independent experts for language and content improvement.

Comments 3 : A lot of statements and claims are missing bibliographic documentation or the references are not primary sources of information.

Response 3 : We sincerely appreciate the reviewers' valuable comments. The references in the manuscript have been thoroughly checked and revised to ensure accuracy and completeness.

Comments 4 : Expressions like "great potential"  and "low immunogenicity" are used throughout the manuscript,  without explanation or specific documentation.

Response 4 : We are grateful for the experts' constructive suggestions. The manuscript now includes enhanced discussion in Section 6 (Conclusion and Outlook), with the following modifications: PDEVs exhibit distinct advantages in clinical translation, particularly in terms of biocompatibility, safety profile, and cross-species applicability. (1) Their inherent biocompatibility and low toxicity stem from significant differences between plant-derived surface proteins and human proteins, which not only minimize immune rejection reactions but also enable repeated administration. This characteristic, combined with the absence of animal virus contamination, ensures superior safety. (2) PDEVs serve as remarkably stable drug delivery platforms, capable of encapsulating diverse therapeutic agents including hydrophilic/hydrophobic drugs, nucleic acids, and natural bioactive molecules. They demonstrate exceptional stability in simulated gastrointestinal environments and maintain integrity through freeze-thaw cycles. (3) Rich in natural bioactive components, PDEVs possess intrinsic targeting capabilities. For instance, ginger-derived EVs loaded with siRNA and chemotherapeutic agents have demonstrated precise delivery to target cancer cells while sparing major organs, concurrently mitigating conventional chemotherapy-induced adverse effects. (4) The scalability of PDEV production is facilitated by their abundant sources and cost-effectiveness, making them particularly suitable for large-scale clinical applications.

Comments 5 : Almost all citations originate from a narrow geographic region. I would suggest that you include landmark international studies and ISEV guidelines.

Response 5 : We sincerely appreciate the reviewers' constructive comments. In response, we have added a new section (Section 5: Clinical Trials of PDEVs) that incorporates data from international clinical trial registries. The characterization and identification of PDEVs performed in accordance with ISEV guidelines.

Comments 6 : The review does not adequately discuss controversies, limitations, or conflicting findings in the field.

Response 6 : We sincerely appreciate the reviewers' insightful comments. In response, we have enhanced Section 6 (Conclusion and Outlook) by incorporating a comprehensive discussion of the current limitations and ongoing controversies surrounding PDEVs. The specific modifications include: Currently, PDEVs exhibit both promise and controversy. Key issues include: (1) Nomenclature and Classification Ambiguity: The terminology for PDEVs remains inconsistent, often overlapping with terms such as "plant nanovesicles (PNVs)" or "plant exosome-like nanoparticles (PELNs)." (2) Lack of Universal Markers and Compositional Heterogeneity: Unlike mammalian exosomes, PDEVs lack well-defined surface markers (e.g., CD9, CD63), and their lipid, protein, and nucleic acid profiles vary significantly across plant species. (3) Unclear Cross-Kingdom Regulatory Mechanisms: Although PDEVs can mediate cross-kingdom gene regulation, their precise molecular targets and signaling pathways require further validation. Additionally, challenges in safety and clinical translation persist, including: Uncertainty in Administration Routes and Pharmacokinetic Stability: The oral bioavailability, gastrointestinal stability, and targeting efficiency of PDEVs remain poorly understood. Limited Clinical Evidence: Most studies are confined to preclinical models, with only a few PDEVs advancing to clinical trials. Addressing these issues is critical for advancing PDEVs toward therapeutic applications.Comments 7 : The authors  mention that PDEVs have similar or superior therapeutic potential compared to MDEVs, but without adequate supporting evidence

Response 7 : Thank you very much for the experts' suggestions. The authors have made revisions and additions to the content in the 6. Conclusion and outlook section. In the second paragraph of this section, the authors have added the limitations faced by MDEVs and compared them with PDEVs, further highlighting the advantages of PDEVs. The revised content is as follows: Compared to MDEVs, PDEVs have significant inherent advantages, while MDEVs face many limitations. First of all, because MDEVs-enriched fluids usually contain multiple impurities and have similar sizes and densities, which lead to difficulties in characterization, better methods need to be developed to better quantify and characterize MDEVs. Secondly, the low yield of EVs is also a challenge. Furthermore, the utilization rate of MDEVs is low, the are easy to be degraded and difficult to function for a long period, therefore, it is necessary to use biomaterials loaded with MDEVs to slow down their. Finally, MDEVs also face immunogenic risks, such as bacterial-derived EVs potentially containing lipopolysaccharides or other pathogen-associated molecules, which can easily trigger a strong immune response or sepsis risk.

Comments 8 : There is no discussion of pharmacokinetics, biodistribution, GMP manufacturing, regulatory challenges, or ongoing clinical trials.

Response 8 : Thank you very much for the expert's comments. I have already supplemented the relevant content in section 6 of the manuscript. In the Conclusion and outlook section, I have supplemented the relevant content and added section 5. Clinical Trials of PDEVs. section. Specific modifications are as follows:

(1)6. Conclusion and outlook. Added Content:Despite their therapeutic potential, research on PDEVs remains in its nascent stage, with several critical challenges requiring resolution: 1. Key Research Gaps: (1) Absence of standardized biomarkers and characterization protocols. (2) There remains a limited understanding of their biodistribution and pharmacokinetic profiles, particularly regarding carrier-drug interactions. Key challenges include: Inefficient targeting. Limited ability to cross biological barriers. Difficulty in achieving an optimal balance between controlled release and systemic clearance. Complex interactions between the carrier and the drug. (3) Limited mechanistic studies on their therapeutic actions. (4) Lack of comprehensive safety evaluation and regulatory frameworks. (5) Technical challenges in scalable isolation and purification methods. 2. Critical Development Needs: (1) Establishment of Good Manufacturing Practice (GMP) standards. (2) Development of quantitative analytical methods. (3) Implementation of quality control parameters (4) Creation of standardized storage and stability protocols. 3. Future Research Priorities: (1) Elucidation of structure-function relationships. (2) Investigation of long-term biological effects. (3) Optimization of cargo loading efficiency. (4) Development of targeted modification strategies. As fundamental research progresses, PDEVs show tremendous promise for revolutionizing regenerative medicine by: (1) Providing novel therapeutic modalities for tissue repair, (2) Enabling targeted drug delivery across biological barriers, (3) Offering cost-effective alternatives to current cell-based therapies.

(2)5. Clinical Trials of PDEVs. Added Content:Although people's enthusiasm is growing day by day, currently only a few clinical trials have begun to explore the characteristics of PDEVs. The first clinical attempt was started in August 2012, and was focused on evaluating the effectiveness of grape-derived EVs in preventing oral mucositis in patients undergoing chemoradiation treatment for head and neck cancer[92]. The second item is a study that focuses on investigating the ability of PDEVs to deliver curcumin to normal and colon cancer tissue. [93] However, as the research is still in its infancy, there are very few clinical trials on PDEVs in progress or completed, See Table 3 for details.

Table 3 Two Clinical Trials Related to PDEVs

PDEVs

Disease

Clinical Phase

Outcome

Literature

Grape-derived EVs

Head and neck cancer

Phase I

Effectiveness of grape-derived EVs in reducing oral mucositis in patients undergoing chemoradiotherapy for head and neck cancer.

[92]

curcumin

Colon cancer tissue

Phase I

Delivery of Curcumin to Normal and Colon Cancer Tissues by PDEVs.

[93]

Round 2

Reviewer 2 Report

Comments and Suggestions for Authors

The paper has been significantly ameliorated. There are still several errors in the English language. Such as 

Figure 5 lines 429-430. Please correct English "in the treatment of promotes" 

Comments on the Quality of English Language

 There are still several errors in the English language. 

Figure 5 lines 429-430. Please correct English "in the treatment of promotes" 

Author Response

Comments : The paper has been significantly ameliorated. There are still several errors in the English language. Such as Figure 5 lines 429-430. Please correct English "in the treatment of promotes"

Response :Thank you very much to the experts for their careful review and comments on this manuscript. The authors have made the necessary revisions according to the requirements and have thoroughly checked the content of the manuscript to ensure that similar errors will not occur. The specific revisions are as follows: “Schematic illustration of the therapeutic mechanisms of PDEVs in promoting wound healing and treating musculoskeletal system diseases. (a) Schematic illustration of the therapeutic mechanisms of PDEVs in promoting wound healing. (b) Schematic illustration of the therapeutic mechanisms of PDEVs in treating musculoskeletal system diseases.”
